# High-Sensitivity Ammonia Sensors with Carbon Nanowall Active Material via Laser-Induced Transfer

**DOI:** 10.3390/nano12162830

**Published:** 2022-08-17

**Authors:** Alexandra Palla-Papavlu, Sorin Vizireanu, Mihaela Filipescu, Thomas Lippert

**Affiliations:** 1Lasers Department, National Institute for Lasers, Plasma and Radiation Physics, Atomiștilor 409, 077125 Măgurele, Romania; 2Laboratory of Inorganic Chemistry, Department of Chemistry and Applied Biosciences, ETH Zurich, 8093 Zurich, Switzerland; 3Laboratory of Multiscale Materials Experiments, Paul Scherrer Institute, 5232 Villigen, Switzerland

**Keywords:** CNW, LIFT, carbon nanowalls, chemiresistor, ammonia, laser transfer

## Abstract

Ammonia sensors with high sensitivity, reproducible response, and low cost are of paramount importance for medicine, i.e., being a biomarker to diagnose lung and renal conditions, and agriculture, given that fertilizer application and livestock manure account for more than 80% of NH_3_ emissions. Thus, in this work, we report the fabrication of ultra-sensitive ammonia sensors by a rapid, efficient, and solvent-free laser-based procedure, i.e., laser-induced forward transfer (LIFT). LIFT has been used to transfer carbon nanowalls (CNWs) onto flexible polyimide substrates pre-patterned with metallic electrodes. The feasibility of LIFT is validated by the excellent performance of the laser-printed CNW-based sensors in detecting different concentrations of NH_3_ in the air, at room temperature. The sensors prepared by LIFT show reversible responses to ammonia when exposed to 20 ppm, whilst at higher NH_3_ concentrations, the responses are quasi-dosimetric. Furthermore, the laser-printed CNW-based sensors have a detection limit as low as 89 ppb and a response time below 10 min for a 20 ppm exposure. In addition, the laser-printed CNW-based sensors are very robust and can withstand more than 200 bending cycles without loss of performance. This work paves the way for the application and integration of laser-based techniques in device fabrication, overcoming the challenges associated with solvent-assisted chemical functionalization.

## 1. Introduction

Currently, the pollutants resulting from continuous industrial development may lead to major disasters, which could affect the equilibrium of ecosystems [1]. Therefore, the detection and monitoring of toxic gases such as ammonia, carbon monoxide, sulfur dioxide, or chloro- and fluor-carbons must be carried out from the moment of their release [2]. In this context, the development of high-sensitivity sensors (low ppm or ppb concentrations) is a hot topic, both in industry and research.

Ammonia is one of the most dangerous gases in industry, due to the fact that it violently reacts with water and produces serious damage to the respiratory system, eyes, and skin [3]. In addition, ammonia represents also a danger to the environment, being a precursor and catalyst in the formation of various fine particles. In particular, fertilizer application and livestock manure account for more than 80% of NH_3_ emissions [4]. The recommendations from the National Institute for Occupational Safety and Health (NIOSH) indicate 25 ppm for 8 h and 35 ppm for 15 min of exposure as the acceptable exposure limits. Thus, NH_3_ detection at low concentrations is a matter of environmental and health concern due to its threshold limit value of 25 ppm.

Thin films and nanostructures play an important role in the development of sensors, considering their multiple uses, i.e., as active materials, transducers for signal detection, or in the architecture of electrical circuits. The interest in nanostructured films comes not only from their reduced dimensionality (which ensures sensor portability and low energy and raw material consumption), but also from their unique physical and chemical properties.

The most common sensors are based on metal–oxide semiconductors (MOSs) (SnO_2_, In_2_O_3_, ZnO, etc.), which have various advantages in terms of high sensitivity, reliability, and relatively small size [5].

However, the main disadvantage is their high operating temperature. For example, SnO_2_ sensors operate at 500 °C, and TiO_2_, CeO_2_, and Nb_2_O are used at >700 °C [6,7], making it challenging to implement them on flexible supports. MOS sensors work by modifying the device’s resistance when gas comes in contact due to the reduction–oxidation processes.

During the last decade, special attention has been devoted to carbon materials having graphene as the base unit (both individual graphene, as well as assembles of several graphene layers) and, recently, to nanostructures based on vertically oriented graphene with various denominations such as: carbon nanowalls (CNWs), nanosheets, nanoflakes, a few layers of graphene, graphene nanowalls, etc. [8,9]. CNWs have a specific morphology and geometry with sharp edges [10], a high specific area, high electronic mobility, and high current densities, which make them very attractive for a wide range of applications including supercapacitors and sensors [11,12,13,14]. For their integration into sensors, CNW-like materials may be considered among the most promising nanoporous/microporous structures, and their surface may exceed a couple of thousand m^2^/g [15]. CNW sensors have been used to detect nitrogen dioxide, acetone, methanol, hydrogen, or ammonia with good sensitivity and reversibility, although their sensitivity decreases over time with multiple uses [16,17,18,19]. Hybrid CNW sensors were also produced by decorating the CNWs with Ag nanoparticles [20] for ammonia detection [21] or with SnO_2_ for formaldehyde detection [22]. To the best of our knowledge, there has been only one report on reversible NH_3_ [20] sensing using chemiresistors based on CNWs decorated with Ag nanoparticles, though the sensor’s response was below 5%. However, the drawbacks of using CNWs in the development of sensors are the high costs associated with the raw materials, the high temperatures that occur during CNW synthesis (usually above 600 °C), and the low adherence to metallic electrodes. In addition, thermal treatments or photo-irradiation are often required to desorb the vapor again for the reversible operation of the sensors [23].

Therefore, the novelty of this study is twofold: (i) we report an efficient strategy for transferring CNW films to flexible substrates for chemiresistor fabrication and (ii) the realization of ultra-sensitive CNW-based flexible sensors for the detection of ammonia concentrations in the low ppb range.

The fabrication strategy for CNW-based chemiresistor sensors is based on laser-induced forward transfer (LIFT) [24]. Briefly, in LIFT, the material of interest, i.e., the donor material, is deposited as a thin film on a laser-transparent substrate. A pulsed laser beam is focused or imaged on the material of interest—transparent substrate interface—and upon interaction, it causes a rapid increase in pressure, which, in turn, leads to the ejection and propelling of a part of the donor onto a substrate placed parallel and in close proximity to the donor. The collected material is named the pixel. This laser-based technique has been extensively used to transfer both solid [25,26,27] and liquid [28] materials from a donor support onto a receiver substrate. The transfer of materials occurs either in air or a vacuum, in a controlled manner, this technique offering high printing resolution and a minimal waste of the material. LIFT has the potential to allow material usage in the range of 80%. LIFT is a clean (solvent-free) and versatile fabrication method, which has already been applied for device fabrication [29,30]. Furthermore, this technique was already shown to transfer carbon nanostructures such as CNTs [31] or even CNWs [32]; therefore, it is a promising method that needs further investigation. More importantly, with this method, it is possible to utilize layer-like structures such as the hybrid CNW nanocomposite, a task that is difficult to achieve with other deposition methods.

In our previous paper [32], we showed that it is possible to print by LIFT CNWs onto polyimide (flexible substrate) and glass (rigid substrate) with good adhesion. The printed pixels remained intact and were not damaged by the laser irradiation, thus proving that LIFT can be used with hybrid CNWs.

In this article, we report for the first time the use of LIFT for the fabrication of CNW ammonia detection sensors on flexible substrates for cost-effective and easy-to-integrate devices. Given the presence of ammonia in the environment either due to natural processes or as a result of industrial activities, including intensive farming, the laser-fabricated CNW-based sensors could be used in various applications, including industrial environmental monitoring and process control or in the pharma industry and laboratories.

## 2. Materials and Methods

### 2.1. Fabrication of CNWs

The synthesis of vertical graphene sheets or carbon nanowall (CNW) layers was carried out by a low-pressure plasma jet [33]. The growth conditions were established following previous studies [34].

Briefly, here, we used an argon plasma jet injected with acetylene and hydrogen at a gas flow ratio of Ar/H_2_/C_2_H_2_ of 1400/25/1 sccm, working pressure ∼130 Pa, 300 W radio frequency power, and 700 °C substrate temperature. To obtain the graphene layers, we used transparent quartz plates (25 mm × 25 mm × 1 mm) as the support. The deposition time was adjusted at 15 min to obtain CNW layers with approximately a 1-micron thickness.

### 2.2. LIFT Setup

The LIFT setup used in this work consisted of a pulsed XeCl laser (Lambda Physik COMPex (Coherent, Santa Clara, CA, USA, 308 nm emission wavelength, 30 ns pulse length, 1 Hz repetition rate), which is guided and imaged with an optical system at the quartz–CNW film (also named the donor) interface to transfer CNW pixels to the collector surface (receiver).

The donor and receiver were placed parallel and at a distance <10 µm onto an *xyz* translation stage from Newport Corporation (Irvine, CA, USA), which allowed the displacement of the donor–receiver system with respect to the laser beam. All experiments were carried out under ambient pressure at temperatures close to room temperature. A scheme of the LIFT setup used in this work is shown in Figure 1. The receiver substrates used for the LIFT experiments were either polyimide foils cut into 25 × 25 × 1 mm pieces or polyimide foils with pre-patterned Pt electrodes used for the functional characterization of the LIFT-printed materials.

The Pt electrodes were sputtered using a 20 nm chromium interlayer with 100 nm of platinum on top (see Figure 1 for the electrode design). The LIFT-printed CNW sensors (similar to the commercial sensors) need a heating step to obtain a stable microstructure, as well as stable sensor behavior. In order to treat the samples, the LIFT-ed sensor pads were heated for 6 h at 150 °C, followed by 6 h at 100 °C in a stream of 1 L/min of synthetic air (SA) containing 20% O_2_ and 80% N_2_.

The transferred pixels, as well as the donor films prior to ablation, were investigated by optical microscopy and scanning electron microscopy (SEM). The optical microscopy images were acquired with a Zeiss Axioplan microscope (Zeiss Group, Oberkochen, Germany) coupled with a ThorLabs digital camera (DCC1545M—USB 2.0 CMOS Camera from Thorlabs Inc. Newton, NJ, United States). The SEM images were acquired with a Zeiss Supra VP55 FE-SEM (Zeiss Group, Oberkochen, Germany) apparatus operating at a voltage of 5 kV and using an in-lens detector.

In addition, micro-Raman spectroscopy was applied in order to investigate the potential heat-induced damage of the CNWs as a result of the laser transfer. Raman spectra prior to and after laser transfer were recorded with a Labram confocal Raman microscope from Jobin Yvon (Horiba Ltd., Kyoto, Japan). The spectra were recorded using the 633 nm HeNe laser as the excitation source, with a power on the samples’ surface of 20 mW. Each of the spectra were collected over the 1000–3500 cm^−1^ range and were acquired after a 20 s exposure time and 5 accumulation times, at room temperature.

### 2.3. Sensor Tests

The performance of the LIFT-printed polyimide foils with pre-patterned Pt electrodes (50 nm thick) (further named sensor pads) was analyzed in a controlled atmosphere, which offered the possibility of adding analytes in the ppm range. The LIFT-printed polyimide foils with pre-patterned Pt electrodes (25 µm wide Pt fingers with 25 µm spacing) were mounted on an alumina block and were contacted electrically by two metal clamps, reaching a total contact resistance of less than 50 Ω. Two-point resistance measurements were acquired by a home-made LabView computer-controlled setup using a Keithley 2400 source meter (Tektronix, Beaverton, OR, USA). The alumina block with the LIFT-printed polyimide foils with pre-patterned Pt electrodes was placed in a tube with a main gas supply of synthetic air (SA) with a standard gas flow of 1 L/min SA.

In order to analyze the response of the LIFT-printed sensor pads to different ammonia concentrations, small amounts, generally 10–50 µL, of liquid ammonia were injected with a gas-chromatography syringe into a balloon (from Carl Roth), which was afterwards filled with dry N_2_, thus achieving a NH_3_ concentration in the 20–100 ppm range. This gas mixture was added with a low flow rate of 0.01 to 0.1 L/min to the main gas flow. After the LIFT-printed sensor pads reached saturation, nitrogen was introduced in the test chamber until the original baseline was recovered.

To study the reliability of the as-fabricated flexible sensors, the transferred CNW sensors were submitted to 200 bending cycles, and their sensing performance towards 20 ppm NH_3_ was recorded at different intervals, i.e., after 100 and 200 cycles, respectively.

## 3. Results

The transfer of an interconnected network of vertically oriented graphene (CNW) with high resolution is of high importance for its integration into sensor devices. In our previous work [32], we used LIFT to deposit CNWs and CNWs decorated with SnO_2_ onto rigid and flexible materials and showed that the best laser fluence for deposition onto a polyimide substrate is 600 mJ/cm^2^ (for a 4 µm thick film).

Here, we optimized the laser transfer process in order to achieve a “clean”, regular transfer of CNW pixels from 1 µm-thick CNW donor layers (an SEM image of a CNW donor is shown in Figure 2). We showed that it is possible to transfer regular CNW pixels for laser fluences of 450 mJ/cm^2^. An array of CWN pixels transferred at a wide range of laser fluences, i.e., from 700 mJ/cm^2^ to 450 mJ/cm^2^, is shown in Figure 3a. The overall appearance of the pixels transferred from 700 mJ/cm^2^ to 550 mJ/cm^2^ is unregular, with large chunks of CNW material around the pixels; only for laser fluences below 500 mJ/cm^2^, a regular transfer with sharp edge pixels is achieved. An SEM image of a CNW donor after LIFT at a 450 mJ/cm^2^ laser fluence is shown in Figure 3b, where it can be clearly seen that, after the laser transfer, the donor remains clean, with no redeposited materials around or inside the ablated pixel (see Figure 3b).

In order to check for any structural or chemical damages resulting from the laser transfer, we carried out micro-Raman spectroscopy on the pixels transferred at 450 mJ/cm^2^ and compared the acquired spectra with the spectra recorded on the donor prior to the transfer (see Figure 4). Both Raman spectra showed the typical Raman peaks of the CNWs, i.e., the peak at 1336 cm^−1^, which is attributed to the defects and disorder due to the phonon coupling on the nanowall edges [14], and the peak at 1598 cm^−1^, which is attributed to the sp^2^ phonon vibration in the CNW layers [35]. In addition, two weaker peaks, i.e., at 2650 cm^−1^, assigned to the overtone of the D band, and at 2900 cm^−1^, assigned to the sum of the D and G bands, can be noticed in both spectra. The Raman investigation demonstrates that the CNWs did not suffer any structural damage as a result of the laser transfer, and thus, the laser-transferred CNW pixels can be further tested in sensing applications.

In order to fabricate CNW-based sensors, we transferred CNW pixels at a 450 mJ/cm^2^ laser fluence onto sensor pads and tested their sensing abilities towards different concentrations of ammonia. The transfers onto the IDT structure did not show significant differences from the transfers onto the polyimide flat substrates. An optical microscopy image of a CNW pixel transferred onto a sensor pad is shown in Figure 5a, while cross-section SEM images taken at a higher magnification of the CNW pixel printed onto the metal electrodes are shown in Figure 5b,c. After the transfer, the as-fabricated sensors were conditioned as described in the experimental section to cure the transferred material.

The images shown in Figure 5 reveal that LIFT transfers onto metal electrodes show a similar behavior as the transfers on even surfaces. To verify whether the transferred pixels are electrically interconnected and suitable for sensing applications, the conductivity of the transferred material was measured. The I–V characteristics of the as-fabricated flexible CNW sensors exhibited an Ohmic behavior (Appendix A).

Given the excellent morphology, structure, stability [11,34,36], and chemistry of the CNWs, i.e., large surface-to-volume ratio, chemical robustness, and well-defined electric properties, we evaluated the chemiresistive response of the laser-printed CNWs towards various concentrations of ammonia, at temperatures close to room temperature (22 ± 1 °C). Prior to evaluating the CNW sensor response to ammonia, we evaluated the baseline resistance and found that it was in the range 2 kΩ–10 kΩ. The NH_3_ sensing properties of the sensors based on LIFT-ed CNW pixels toward different concentrations of NH_3_ are shown in Figure 6a. During the measurements, the temperature T was continuously measured and for the CNWs was T = 22 ± 1 °C. We noticed that when ammonia was introduced in the testing chamber, the resistance of the sensor increased, which is consistent with a p-type semiconductor response, with holes as the majority of carriers (see Figure 6a). Briefly, when the CNWs were exposed to reducing molecules such as NH_3_, the accumulation region at the surface of the CNWs was reduced, which in turn led to a reduction of the hole current and, thus, an increase of the resistance.

Furthermore, we can see that the laser-printed CNW sensor response upon exposure to NH_3_ (N_2_ balance gas) for 30 min was characterized by a fast increase in a short time and a slower decrease upon removal of the NH_3_. The sensor response is defined as ΔR/R_0_ = (R_g_−R_0_)/R_0_, where R_g_ is the resistance upon NH_3_ exposure and R_0_ is the baseline resistance before exposure to NH_3_. In order to quantitatively correlate the CNW sensor responses as a function of the NH_3_ concentrations tested, we plotted the sensor response after 30 min of exposure to different analyte concentrations. Studying the sensor response at different ammonia concentrations (Figure 6b), we found an excellent linear sensor response−analyte concentration relationship (R^2^ = 0.99) for the range of tested concentrations (i.e., 20−100 ppm NH_3_). In addition, in order to quickly assess the sensing performance of the CNW-based sensors at different time intervals upon NH_3_ exposure, we plotted the sensor response versus the analyte concentration, and the linear relationship was maintained also for different exposure times (i.e., from 1.5 min to 30 min) (see Figure 6c). The limit of detection (LOD) for NH_3_ was determined to be 89 ppb for 30 min of exposure and was calculated as previously reported [37,38] from the signal–noise relationship, taking into account the slope (0.0026) of the calibration curve shown in Figure 6b. This LOD value is one order of magnitude lower than that of chemiresistive devices based on CNWs reported previously [19]. We can assume that the superior LOD achieved with the laser-printed CNW-based sensors could be attributed to the high velocities generated in LIFT (around 200 m/s), which led to a firm electrical contact between the metal electrodes and the pixels, allowing for a better draining of the injected carriers through the electrodes [31,37]. Representative examples of NH_3_ detection by CNWs and CNW hybrids, together with their performance parameters (where available) are presented in Table 1.

Generally, when fabricating sensors, the reversibility of the sensor response upon multiple exposures to analytes is a very important and desired property. However, although dosimetric sensors show irreversible responses upon exposure to analytes, they could be useful due to the fact that they can achieve the total exposure cycles [40]. Thus, in order to obtain more information on the performance of the laser-fabricated CNW sensors, we evaluated the initial rate of sensor responses (the first minute after exposure to NH_3_) together with the response and recovery times of the laser-printed CNW-based sensors. The initial response rate of the laser-fabricated CNW sensors gives valuable information on the performance of the sensor, as, at the very beginning of NH_3_ exposure, the ammonia concentration is much lower than the density of active sites on the active CNW surface. Thus, the analyte–CNW surface interaction can be seen as a pseudo-first-order reaction [40,41,42]. Moreover, the analysis of the initial rate of response led us to the conclusion that the CNWs are sensitive to ammonia within only 1 min of the initial exposure, as shown in Figure 6d, and by plotting the slope of the fitting as a function of the ammonia concentration, we found an excellent linear response (see Figure 6e).

Although the sensors display a resistivity change, the recovery appears to be much slower. Thus, we evaluated the response time of the CNW sensors, which is defined as the time the sensors’ resistance reaches 90% of its final value upon exposure to NH_3_. The response times of the laser-printed CNW sensors are shown in Figure 6f. Although the response (for 30 min exposure to the analyte) is in the minutes range, i.e., about 10 min for 20 ppm of NH_3_, this is similar to what was reported previously in [21]. Ultimately, the full reversibility of the laser-printed CNW-based sensors was found only for a 20 ppm ammonia exposure (Appendix A), while the response of the laser-printed CNW-based sensors is partially reversible in the concentration range from 30–80 ppm of NH_3_; thus, the sensors demonstrated a quasi-dosimetric response.

The results shown above indicate that the laser-printed CNWs are useful materials to detect ammonia below its acceptable exposure limit. In addition, the integration of CNWs onto flexible substrates opens up new opportunities for flexible sensing devices or wearable sensors. In order to verify the potential of the laser-printed CNW-based sensors, we tested the sensors with 20 ppm of ammonia after carrying out multiple bending cycles, i.e., 100 and 200. We noticed that without any bending, the initial resistance of the fabricated flexible chemiresistor was measured to be around 6300 Ω at 22 °C. It was observed that the Ohmic behavior did not change after the applied bending cycles (see Figure 7b). In addition, the resistance variation of the chemiresistor at a 20 ppm concentration of NH_3_ was small (~3%) with increasing bending cycles, as shown in Figure 7b. Further studies are underway to evaluate whether this resistance variation is due to the metallic electrodes (i.e., loss of adherence or “wear”) or to the transferred CNW pixels.

To sum up, we may conclude that the practical implications of the sensors developed in this work are highly promising and open up new avenues for printed flexible electronics.

## 4. Conclusions

In summary, we successfully demonstrated the “clean“, solvent-free printing with a high resolution of CNWs onto specific metallic geometries designed on flexible substrates. In the implementation of CNWs as an active material for chemiresistive devices, the CNW pixels showed excellent responses towards NH_3_ with an ultralow LOD, i.e., of only 89 ppb, after 30 min of exposure to ammonia, a value that is superior to the most sensitive sensors based on CNWs reported in the literature. Furthermore, the laser-printed CNW-based sensors are robust and can withstand more than 200 bending cycles without loss of sensitivity towards ammonia.

We believe that this study forms the foundation for the future design and development of CNW-based sensors, paving the path for their usage in chemical sensing, electronics, or electrical energy conversion and storage. Future studies will focus on the exploration of the effect of temperature and/or UV light on the recovery of the sensors, as well as the examination of the influence of different dopants.

## Figures and Tables

**Figure 1 nanomaterials-12-02830-f001:**
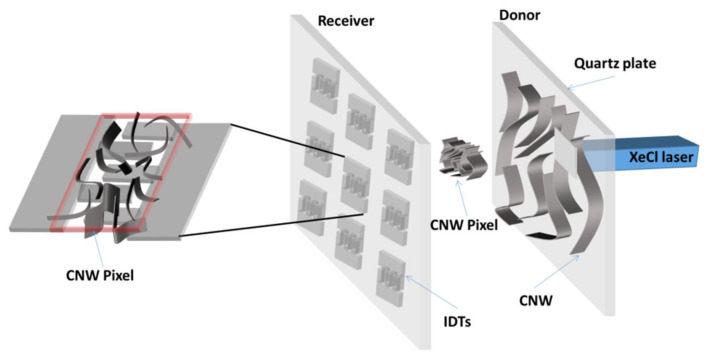
Sketch of the LIFT setup showing the transfer of the CNW donor material onto the receiver substrate (polyimide foil with pre-patterned Pt electrodes). The donor and receiver are placed onto an *xyz* translation stage. The laser beam impinges through the transparent quartz plate and pushes forward a small portion of the CNW film onto the polyimide foil with pre-patterned Pt electrodes.

**Figure 2 nanomaterials-12-02830-f002:**
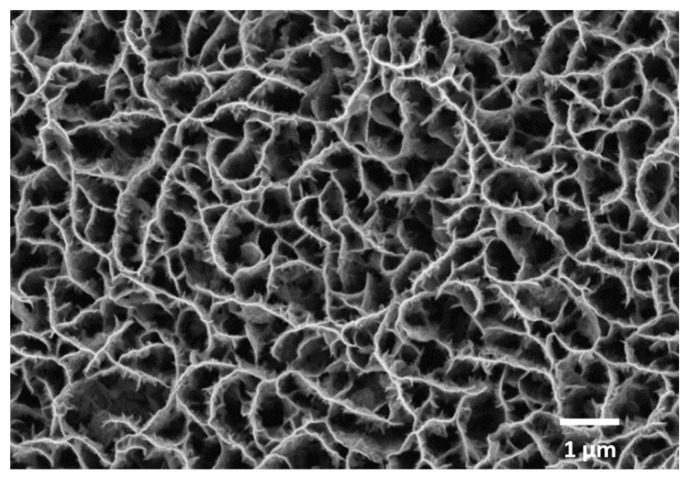
Top-view SEM image of a CNW donor prior to laser-induced forward transfer.

**Figure 3 nanomaterials-12-02830-f003:**
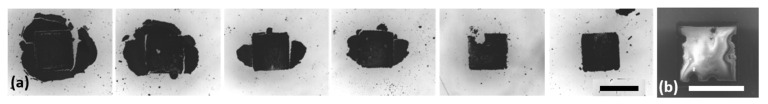
(**a**) CNW pixels transferred at different laser fluences, i.e., from left to right: 700, 650, 600, 550, 500, and 450 mJ/cm^2^. (**b**) SEM image of a CNW donor after LIFT at a 450 mJ/cm^2^ laser fluence. The scale bar in both images, i.e., (**a**,**b**), is 500 µm.

**Figure 4 nanomaterials-12-02830-f004:**
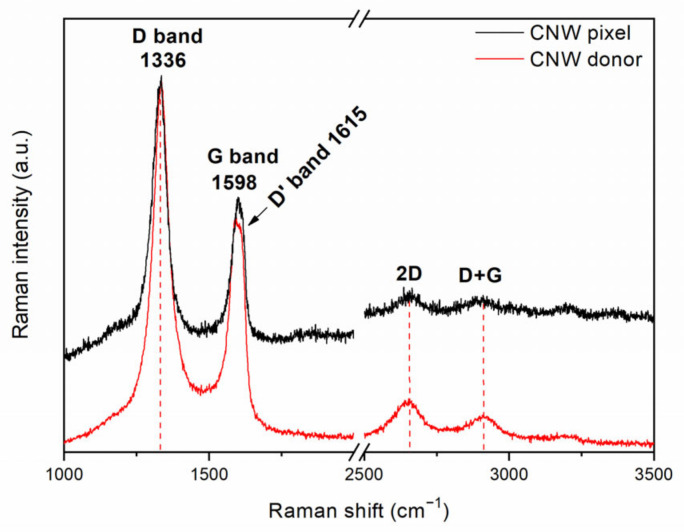
Raman spectra of a CNW donor prior to LIFT and a CNW pixel.

**Figure 5 nanomaterials-12-02830-f005:**
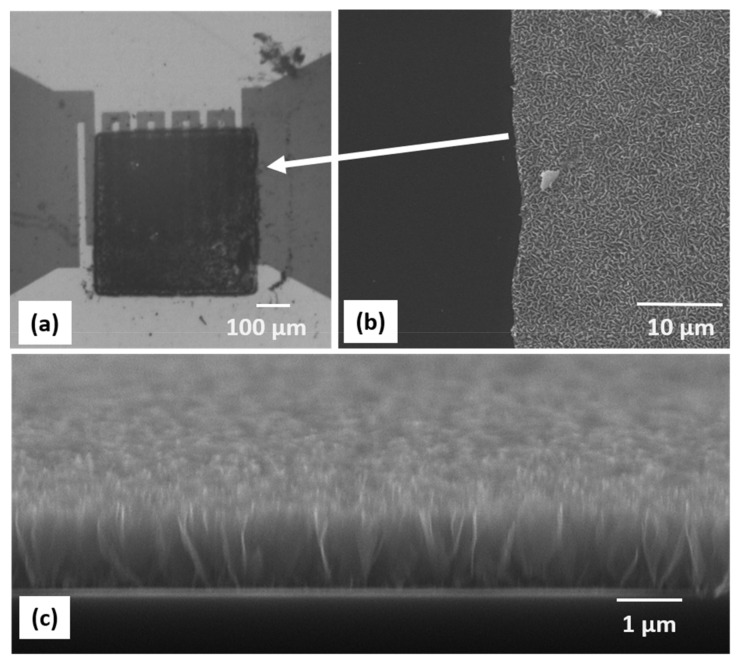
(**a**) Optical microscopy image of a CNW pixel printed onto the IDTs of a chemiresistive sensor. (**b**) Top-view SEM image taken at the edge of the CNW pixel, where a clean-cut, sharp edge can be noticed. (**c**) Cross-section SEM image of a CNW pixel (the pixel thickness is 1 µm).

**Figure 6 nanomaterials-12-02830-f006:**
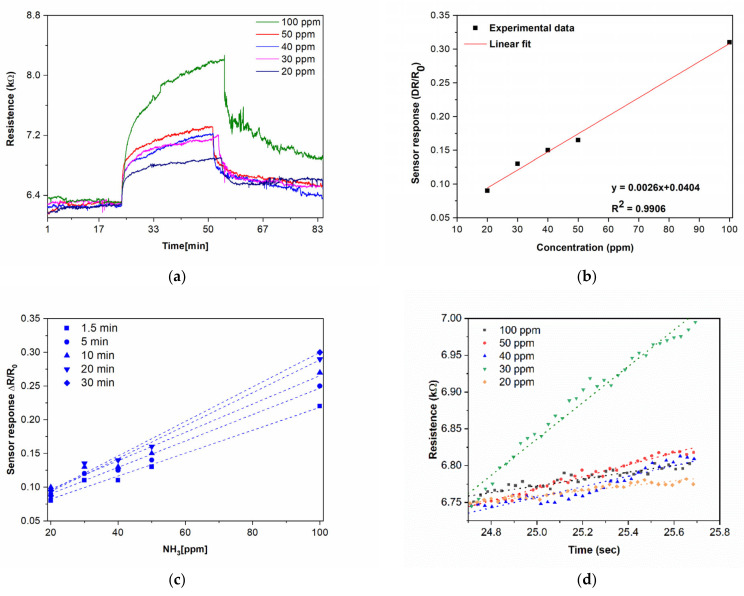
(**a**) Real-time measurements of a CNW sensor printed by LIFT collected for different concentrations, i.e., 20–100 ppm of NH_3_. (**b**) CNW sensor response as a function of NH_3_ concentration at room temperature in N_2_ depicting a good linear relation. (**c**) Response values of a laser-printed CNW-based sensor after 1.5, 5, 10, 20, and 30 min of exposure to NH_3_. (**d**) Responses after 1 min of exposure of the laser-printed CNW-based sensor to different concentrations of NH_3_, i.e., the 20–100 ppm range, and the linear fit of the response. (**e**) The slope of the fit as a function of NH_3_ concentration. (**f**) Response times for the laser-printed CNW-based sensor exposed to different concentrations of NH_3_ (20–100 ppm).

**Figure 7 nanomaterials-12-02830-f007:**
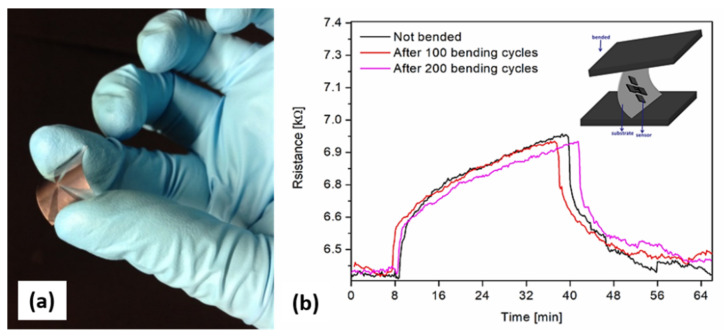
(**a**) Image of a flexible laser-printed CNW-based sensor. (**b**) Real-time measurement of a flexible laser-printed CNW-based sensor to 20 ppm of NH_2_ without any bending and after 100 and 200 bending cycles.

**Table 1 nanomaterials-12-02830-t001:** Representative examples of NH_3_ detection by CNWs and CNW hybrids, together with their performance parameters.

Material	Analyte	Concentration (ppm)	Sensor Performance	Ref.
Epitaxial graphene nanowalls	H_2_	0.5–500	LOD 0.5 ppm	[18]
			500 ms response time	
CNWs	NH_3_	1% in air	NA *	[14]
	NO_2_	100 ppm	NA	
CNWs	NH_3_	1% in air	NA	[39]
	NO_2_	100 ppm	NA	
CNWs	NH_3_ NO_2_	25–100 25–100	LOD 15 ppm LOD 20 ppm	[19]
CNWs with 100 and 300 nm interwall distance	Acetone	NA	Response time of (100 nm) 65.4 (300 nm) and 8.5	[16]
	Methanol Diethyl ether Iso–pentane	NA NA NA	(100 nm) 327.1 and (300 nm) 36.2 (100 nm) 273.8 and (300 nm) 34.1 (100 nm) 87.2 and (300 nm) 46.6	
Vertical graphene nano–petals	NH_3_	10–100 ppt	Joule heating to 100 °C is used during desorption	[21]

* NA = not available.

## Data Availability

The data used to support the findings of this study are available from the corresponding author upon request.

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
