# Peer review of "High-Sensitivity Ammonia Sensors with Carbon Nanowall Active Material via Laser-Induced Transfer"

_nanomaterials, 2022, doi:10.3390/nano12162830_

Round 1

Reviewer 1 Report

The research article on " High sensitivity ammonia sensors with carbon nanowalls active material via laser-induced transfer " reports the fabrication of ultra-sensitive ammonia sensors by laser-induced forward transfer. Below are some of the comments that authors should look into improving the manuscript.

  1. The authors should clearly state the novelty of the work.
  2. It would be better if the authors could provide more information about the linear range of detection.
  3.  After several bending cycles, the resistance gets dropped. What could be the reason?
  4. How stable are the CNW materials as the sensor? As material stability is an important factor for sensors, it would be better if the authors provided additional information on sensor stability.

Author Response

Dear Editor,

Dear Reviewers,

Please, find here attached the revised manuscript “High sensitivity ammonia sensors with carbon nanowalls active material via laser-induced transfer" by A. Palla-Papavlu, S. Vizireanu, M. Filipescu, and T. Lippert submitted for publication at the Nanomaterials journal. On behalf of all the authors, I would like to thank you and the reviewers for the time and effort devoted to evaluate our manuscript “High sensitivity ammonia sensors with carbon nanowalls active material via laser-induced transfer”. We appreciate the overall positive comments by the reviewers and try to comply and answer their requests.

We are looking forward to your response.

Sincerely yours,

Alexandra Palla-Papavlu

Reviewer 2 Report

The authors prepared carbon-nanowall(CNW)-based sensors for ammonia detection using the laser-induced forward transfer (LIFT) method. The evaluation of the sensing performance showed a low limit of detection (89 ppb) and durability against bending cycles. I think that the manuscript can be published after minor revision.

1. Do the black and white bars in Figure 3 indicate the sizes of CNW pixels? If so, please add the values of the lengths shown by the bars in the figure or the caption.

2. Figure 4: “Raman shift” is often used for the x-axis label of Raman spectra.

3. Lines 217 and 219: I think that not Figure 4 but Figure 5 should be referred to here. And Figure 5b looks like a magnified SEM image of the red-squared area in Figure 5a. Please redraw them to clarify that Figure 5b is the cross-sectional SEM image.

4. In lines 225-227, the thicknesses of electrodes and the CNW pixels are discussed, however, their values of thicknesses are not given. Please add them. Furthermore, please show a cross-sectional SEM image exhibiting the thickness of the CNW pixels.

5. Have the authors evaluated the repeatability of the laser-printed CNW sensors for the detection of ammonia gas? Does the sensor show any response to the second or later exposure to ammonia?

6. Lines 295-296: It is not possible to determine whether the sensors exhibit ohmic behavior or not from Figure 7b. Please add their I-V curves.

7. Please add the discussion of why the laser-printed CNW sensors can achieve a superior LOD value than other CNW-based sensors.

8. Please unify the style of references.

Author Response

(The authors gave the same response as above.)

Reviewer 3 Report

The article: High sensitivity ammonia sensors with carbon nanowalls active material via laser-induced transfer was peer-reviewed and publication of the paper is proposed after minor revision. The paper is very well structured and will be of interest to many readers of Nanomaterials. The only problem I find is that no interference study is presented, i.e., can other gasses besides ammonia affect sensor response?

Author Response

(The authors gave the same response as above.)

Reviewer 4 Report

The presented paper fulfills the Journal Scopus. This article is based on 41 articles in the literature over the last thirty years. The results are clearly stated. The authors, motivated by cited literature, carried out experiments and confirmed known knowledge. The work presents new possibilities for obtaining sensor materials on flexible substrates. The results presented are part of the development of the discipline.

Recommendation Regarding This Manuscript: Accept in the present form

Author Response

(The authors gave the same response as above.)

Round 2

Reviewer 1 Report

The revision looks better and satisfactory